# A Generalized Network for MRI Intensity Normalization.

**Attila Simkó**[1]                                                                              ATTILA.SIMKO@UMU.SE
**Tommy Löfstedt**[1]                                                                      TOMMY.LOFSTEDT@UMU.SE
**Anders Garpebring**[1]                                                             ANDERS.GARPEBRING@UMU.SE
**Tufve Nyholm**[1]                                                                          TUFVE.NYHOLM@UMU.SE
**Joakim Jonsson**[1]                                                           JOAKIM.JONSSON@RADFYS.UMU.SE
[1] *Department of Radiation Sciences, Umeå University, Umeå, Sweden*

**Editors:** Under Review for MIDL 2019

## Abstract

Image normalization, the correction for intra-volume inhomogeneities in magnetic resonance imaging (MRI) data has little significance for visual diagnosis, but is a crucial step before automated radiotherapy solutions. There are several well-established normalization methods, however they are usually time expensive and difficult to tune for a specific dataset. In this study, we show how an artificial neural network (ANN) can be trained on non-medical images—making the model general—for intensity normalization on medical MRI images. Compared to one of the most well-known correction methods, N4ITK, the trained network achieves a higher accuracy with a speedup-factor of almost 70.

**Keywords:** Intensity normalization, gain field, magnetic resonance imaging, artificial neural network, machine learning.

## 1. Introduction

The signal intensity of homogeneous tissue from Magnetic Resonance Imaging (MRI) data is seldom homogeneous. The combination of the image acquisition process (*e.g.*, inhomogeneous transmit/receive $B_1$ field (McRobbie et al., 2006)) and the patient anatomy (*e.g.*, tissue-specific radio frequency penetration (Belaroussi et al., 2006)) creates an intra-volume inhomogeneity with no anatomical relevance. Their collective effect is named a gain field, causing a 10 % to 40 % intra-volume, smooth low-intensity intensity variation.

Although having little impact on visual diagnosis, intensity normalization, which is correction for the gain field, is a crucial pre-processing step for automated radiotherapy solutions. A detailed study of gain fields (Sled et al., 1998) and its following proposal of the N3 correction, was later improved by perhaps the most well-known and currently most commonly used method, the N4ITK (Tustison et al., 2010). However, the N4ITK method requires some unintuitive parameter tuning in practice, and is usually time expensive.

There is a concern in medical applications regarding the ability of ANNs to generalize to new data. For a normalization method based on ANNs to be relevant for practical use, it must perform equally well regardless of the scanner parameters and the scanned region of the body. The goal of this work is to achieve generalization by discarding medical images from the training process. The resulting network produces results that rival the N4ITK (with optimized parameters) in both time and accuracy.

## 2. Method

### 2.1. Data

The training dataset was based on a random subset of ImageNet[1], covering thousands of different objects, but no medical images. A total of 108,000 images were cropped to $256 \times 256$, converted to grayscale, and normalized to the range $[0, 1]$.

The test dataset was based on a collection of 20 separate synthetic tissue map volumes, from the online Simulated Brain Database, available from BrainWeb[2]. 12,000 $T_1$-weighted MRI brain scans were simulated using an in-house MR-simulator on all three planes of size $256 \times 256$ with the corresponding tissue maps saved for later evaluation.

Based on the non-uniform fields provided by BrainWeb and literature on the characteristics of gain fields (Hou, 2006; Vovk et al., 2007), a *Gain Field Generator* was created mimicking this behavior. A randomly generated gain field with a maximum 30 % inhomogeneity was added to all training and test samples.

### 2.2. Proposed Architecture

The formation of an image, $v$, can be described by

$$v = u \odot g + n, \tag{1}$$

where $u$ is the true spatial distribution of the signal intensity that only contains intensity variations of relevance, $g$ is a low-frequency multiplicative gain field, and $n$ is an additive Gaussian noise. In "natural" images, the gain field, $g$, can be caused by changes in lighting, or by a smooth color gradient or fading. The operator $\odot$ denotes element-wise multiplication, and $\oslash$ denotes element-wise division.

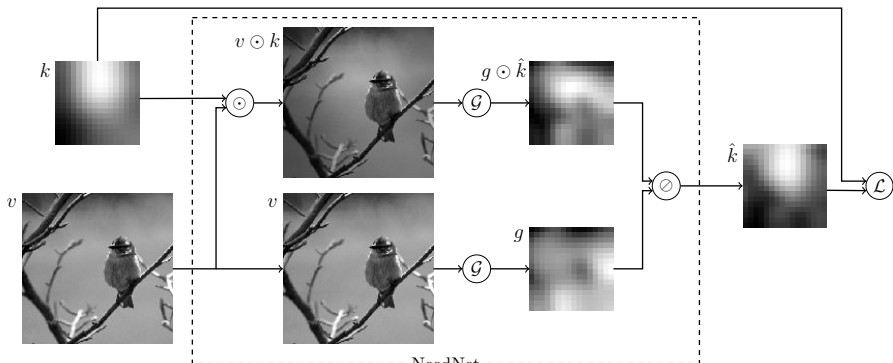

Figure 1: An illustration of the model. The input is a generated gain field and an image (from ImageNet). The parts are denoted as in Equation 3.

We used a modified version of ResNet (He et al., 2015)—that gets us the low-frequency gain field (denoted *GetNet*) of size $16 \times 16$ from an input image of size $256 \times 256$, such that

$$\mathcal{G}(v) = g. \tag{2}$$

---

1. http://www.image-net.org/
2. http://brainweb.bic.mni.mcgill.ca/

Table 1: The relative MAE between Original and Gain, and between Original and the corrected images, together with the computational time of the correction methods.

| Relative MAE | **Gain** | **N4ITK** | **GetNet** |
| --- | --- | --- | --- |
| Entire image | 0.231 | 0.166 | 0.152 |
| Cerebrospinal fluid | 0.212 | 0.083 | 0.049 |
| Gray Matter | 0.215 | 0.065 | 0.031 |
| White Matter | 0.217 | 0.057 | 0.019 |
| Time (s) | — | 20,470 | 294 |

A second network was created that defines what the output of GetNet needs to be (denoted *NeedNet*). The pipeline for training is illustrated in Figure 1. It uses two instances of GetNet, one fed with the training image $v$ and an added gain field $k$, and one fed with only the input $v$. The NeedNet is then defined as

$$\mathcal{N}(v, k) = \mathcal{G}(v \odot k) \oslash \mathcal{G}(v) = \mathcal{G}(u \odot (g \odot k) + n \odot k) \oslash \mathcal{G}(u \odot g + n) \equiv \hat{k}, \qquad (3)$$

where we note that $\mathcal{N}(v, k) \approx (g \odot k) \oslash g = k$. Hence, assuming that $k$ is has similar statistics as $g$, knowing $g$ becomes unnecessary, since we can train the network to discard anything that exists on both images. Finally, for training we used the mean absolute error (MAE) loss, $\mathcal{L}(k, \mathcal{N}(v, k)) = \frac{1}{n} \|k - \hat{k}\|_1$, for $n$ the number of voxels in $k$.

## 3. Results and Discussion

A relative MAE was computed between the input image (Original) and the three versions: Original with an added gain field (Gain), Gain corrected by N4ITK (with optimized parameters), and Gain corrected by GetNet. The results are presented in Table 1.

Both the relative MAE, presented in Table 1, and visual comparisons (see example in Figure 2) show the improvement in accuracy when using GetNet compared to N4ITK, with a speedup-factor of almost 70, using a GeForce GTX 1050 Ti. Both corrections achieve a lower relative MAE for the tissue-specific evaluation, and here the improvements when using GetNet are even more significant.

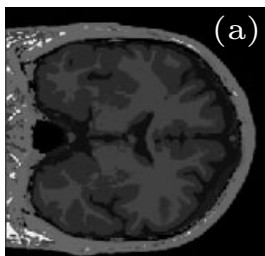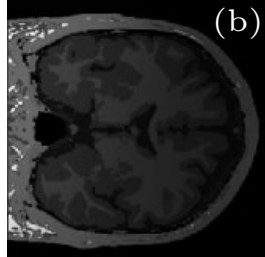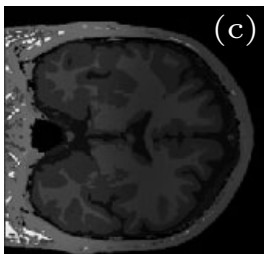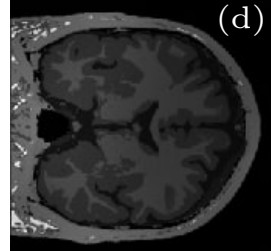

Figure 2: Test dataset example. (a) Original, (b) Gain, (c) N4ITK, and (d) GetNet.

## Acknowledgments

We are grateful for the financial support obtained from the Cancer Research Foundation in Northern Sweden and from Karin and Krister Olsson. This research was conducted using the resources of the High Performance Computing Center North (HPC2N).

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
