# OpenReview forum: "A Generalized Network for MRI Intensity Normalization."
_MIDL.io/2019/Conference/Abstract — MIDL Abstract 2019_

### Official Review · AnonReviewer2 · 2019-04-29
**Abstract Review**

**Rating:** 2
**Confidence:** 2

**Review:**

The authors propose a normalization strategy based on neural networks, which outperforms the N4ITK method both in terms of Mean absolute error and competition time.

Pros

- Image normalization is an important step for complete automatization of image analysis pipelines

Cons

- The proposed method is not well-explained. The utilized Res-net architecture is not clear. I strongly suggest the authors to write and/or illustrate the specific components of “Getnet” and “Neednet”
- The abstract is written with poor English and very difficult to follow the main idea for the reader.

Minor issues

- Please explain with a clear sentence the loss function of the network and its relevance to image normalization.
- Please correct in abstract “Speedup” → speed up
- Please correct “assuming that k is has similar”
- Please specify the term here in the last sentence of the abstract: “ and here the improvements when using GetNet are even more significant”

---

### Official Review · AnonReviewer1 · 2019-05-01
**a relevant and practical solution for an important problem**

**Rating:** 3
**Confidence:** 2

**Review:**

Overall, there is a relevant and practical solution for an important problem. Good speed-up and with a higher accuracy are plus.

---

### Decision · Program_Chairs · 2019-05-06
**Acceptance Decision**

Accept